DNA metabarcoding of mites from small soil samples: limited agreement with morphological identifications but improved results from long-read sequencing

Varusk Sirle 1
Sammet Kaarel 2
Ariyan Manikandan 1
Mubarak Alwutayd Khairiah 3
Anslan Sten sten.anslan@gmail.com 1 3 4
1 Institute of Ecology and Earth Sciences, University of Tartu , Tartu , Estonia
2 Chair of Biodiversity and Nature Tourism, Estonian University of Life Sciences , Tartu , Estonia
3 Department of Biology, College of Science, Princess Nourah bint Abdulrahman University , Riyadh , Saudi Arabia
4 Department of Biological and Environmental Science, University of Jyväskylä , Jyväskylä , Finland
Brygadyrenko Viktor
Electronic publication date: 2025 Oct 20
Publication date: 2025
Volume: 13
Electronic Location ID: e20205
Received 2025 Jul 8; Accepted 2025 Sep 17
Copyright: ©2025 Varusk et al.
Copyright year: 2025
Copyright holder: Varusk et al.
License: This is an open access article distributed under the terms of the Creative Commons Attribution License, which permits unrestricted use, distribution, reproduction and adaptation in any medium and for any purpose provided that it is properly attributed. For attribution, the original author(s), title, publication source (PeerJ) and either DOI or URL of the article must be cited.
License URL: https://creativecommons.org/licenses/by/4.0/

Keywords: Soil mites, PacBio, Illumina, Metabarcoding, Soil fauna

Funding: Researchers Supporting Project, Princess Nourah bint Abdulrahman University HCPNU2025R402 European Regional Development Fund The Program Mobilitas Pluss MOBTP198 Research Council of Finland 362828 This work was supported by the Researchers Supporting Project (HCPNU2025R402) at Princess Nourah bint Abdulrahman University (Riyadh, Saudi Arabia), European Regional Development Fund and the program Mobilitas Pluss (MOBTP198) and by the Research Council of Finland (Decision number 362828). The funders had no role in study design, data collection and analysis, decision to publish, or preparation of the manuscript.

==============================
The characterization of soil mite (Acari) communities traditionally follows morphological identifications of specimens extracted from soil, which is a highly laborious and time-consuming process. Metabarcoding has become an increasingly utilized approach for species identification from environmental DNA (eDNA) samples, but whether the metabarcoding approaches align with the morphological identification data on soil mites has rarely been addressed. Here, we examine the congruence of soil mite communities between morphological and metabarcoding datasets. The morphological dataset was generated by extracting mite specimens from the soil samples, whereas molecular datasets represent two types of cytochrome c oxidase subunit I (COI) amplicons produced directly from soil eDNA (from 0.2 g and 2 g soil samples) and sequenced with Illumina (313 base pairs amplicons) and PacBio (658 base pairs amplicons) platforms. We found that specimen extraction from soil samples, followed by morphological identification, yielded the highest number of mite species. Despite significantly lower mite richness in the metabarcoding datasets, PacBio datasets provided more reliable community profiles that aligned strongly with the morphological data. This indicates that soil sample quantities generally used for microbial analyses are also informative in studying soil faunal communities. Furthermore, our results indicate that methodological choices (herein PacBio vs. Illumina) have a greater influence on mite community detection than the amount of input soil used for DNA extraction. Interestingly, the patterns of the entire metazoan community in the metabarcoding datasets strongly mirrored those of the morphologically identified mite communities alone, indicating that soil mites serve as a powerful ecological indicator group.

Introduction

Soil mites (Acari) are among the most diverse and ecologically significant groups of microarthropods in terrestrial ecosystems, playing crucial roles in decomposition, nutrient cycling, and soil structure maintenance (Coleman, Geisen & Wall, 2024). Due to their sensitivity to environmental changes, they are also widely used as bioindicators (Breure et al., 2005). Traditionally, the identification of soil mites has relied on morphological examination, which requires considerable taxonomic expertise and is often time-consuming due to the small size and morphological similarity of many mite taxa. Morphological identification of specimens is further complicated by the presence of juvenile stages, which frequently lack the diagnostic features for reliable species-level identification (Coleman, Callaham & Crossley Jr, 2017).

Molecular approaches, particularly DNA metabarcoding, has emerged as a powerful tool for biodiversity assessment. Metabarcoding has the potential to overcome several limitations of morphological identification, offering increasing throughput, the ability to detect cryptic and juvenile forms, and the possibility of standardizing biodiversity assessments across studies (Taberlet et al., 2018). By comparing morphological with metabarcoding-based identification of soil arthropods, Oliverio et al. (2018) and Ustinova et al. (2021) have reported correlating richness and community composition patterns among methods. Similarly, Young & Hebert (2022) found congruent diversity patterns for soil arthropods between soil eDNA and bulk arthropod samples. Therefore, metabarcoding approaches have been further applied in several studies for identifying the composition of arthropods in the soil (Andujar et al., 2022; Arribas et al., 2016; Perdomo-González et al., 2025; Sahdra et al., 2025).

However, soil arthropod metabarcoding studies have utilized second-generation sequencing technologies for sequencing short reads (158–418 base pairs; Oliverio et al., 2018; Andujar et al., 2022). Second-generation sequencing technologies, such as Illumina, allow for sequencing of hundreds of millions of reads with high accuracy, but are limited to read lengths of 2 × 300 base pairs. In contrast, third-generation sequencing platforms, such as PacBio, can generate longer reads, allowing more accurate taxa classifications (Tedersoo et al., 2021). Recent advances in third-generation sequencing technologies have substantially improved their accuracy, generating growing interest in metabarcoding approaches that target full-length DNA barcodes (Doorenspleet et al., 2025; Latz et al., 2022; Srivathsan et al., 2024), which typically exceed the read length limitations of second-generation platforms. Additionally, although previous studies have shown that morphological and metabarcoding datasets of soil arthropods may yield correlated results, it remains unclear how the quantity of soil used for DNA extraction influences this relationship.

The objective of this study was to determine whether the identification of mite communities from soil samples using short-read (Illumina) and long-read (PacBio) metabarcoding aligns with results from morphological identification. To this end, we used two commonly applied COI primer pairs: mlCOIintF/jgHCO2198 (Geller et al., 2013; Leray et al., 2013) to amplify a ∼313 bp fragment for Illumina sequencing, and LCO1490/HCO2198 (Folmer et al., 1994) to amplify a ∼658 bp fragment for PacBio sequencing. We further explored how the amount of soil (0.2 g and 2 g) used for DNA extraction influences mite and overall metazoan community detection in metabarcoding approaches.

Materials & Methods

Soil samples (including litter) were collected from five sites between July and September 2022 (Table 1). Six samples were taken from each site along a north-south transect, spaced three meters apart. The size of each sample collection site was 10 × 20 cm, with a depth of five cm. The collected material per sample was placed in zip-lock bags, then homogenized (thoroughly mixed in between hands) and divided into two equal ∼0.5 L parts. One part was used to extract soil animals with Tullgren funnel method for morphological identification, and other for metabarcoding. On the day of the sample collection, one sample replicate was subjected to Tullgren funnel heat extraction (7 days; 95% ethanol in the collection tubes), and the other replicate was dried in a clean paper bag in a drying cabinet with active airflow at 38 °C for at least 24 h (for metabarcoding).

Morphological identification

After Tullgren funnel extraction, mites were sorted and counted using a Leica S8APO stereomicroscope at 60× magnification. Up to six adult individuals per sample from each morphotype were selected for morphological identification. The specimens preserved in ethanol were first transferred to distilled water for 5 min following incubation in a lactic acid at approximately 60 °C for 10 min (with exceptions of adult female Mesostigmata, Prostigmata, hypopi of Astigmata and smaller Oribatid mites, which were left in lactic acid for 24 h). After incubation, mites were studied in cavity slides or rinsed in distilled water for 5 min and then mounted on permanent slides using Hoyer’s medium. The slides were examined using a Leica DM 6000 P microscope (Germany) and, in some cases, also with a Zeiss EVO MA 15 scanning electron microscope to confirm the identifications (Germany). Mites were identified either to species (Oribatida and Mesostigmata) or to higher taxonomic levels (Prostigmata and Astigmata) according to (Hernandes et al., 2016; Karg, 1989; Karg, 1993; Khaustov, 2008; Krantz & Walter, 2009; Mašán, 2003; Weigmann, 2006). Followingly, we refer to the morphological dataset as mites identified morphologically after the Tullgren funnel extraction.

Table 1 Sampling sites in this study.

The “Morph” treatment indicates that mites were extracted from soil samples for morphological identification. The “DNA (0.2 g, 2 g)” treatment refers to samples used for metabarcoding, where DNA was extracted from both 0.2 g and 2 g soil subsamples.

Site	Coordinates	Date	Treatment	Site description	
Site 1
(6 samples)	58.3442°N 26.684°E	15.07.2022	Morph, DNA (0.2 g, 2g)	Mixed deciduous forest, dominated by Betula pendula	
Site 2
(5 samples)	58.3447°N 26.6856°E	15.07.2022	Morph, DNA (0.2 g, 2g)	Mixed deciduous forest, dominated by Salix sp.	
Site 3
(6 samples)	58.3027°N 26.5486°E	09.08.2022	Morph, DNA (0.2 g, 2g)	Mixed deciduous forest, dominated by Populus tremula	
Site 4
(6 samples)	58.3017°N 26.5532°E	09.08.2022	DNA (0.2 g, 2g)	Young mixed deciduous forest, dominated by Betula pendula	
Site 5
(6 samples)	58.3934°N 26.6963°E	01.09.2022	DNA (0.2 g, 2g)	Alvar meadow	

Metabarcoding

The dry soil samples were transferred into zip-lock bags and homogenized by hand to a fine powder. From each sample, two subsamples, 0.2 g and 2 g, were weighed for DNA extraction. DNA was extracted using the DNeasy PowerSoil Pro Kit (0.2 g; Qiagen, Hilden, Germany) and the PowerMax Soil Kit (2 g; Qiagen, Hilden, Germany) according to the manufacturer’s instructions.

Polymerase chain reaction (PCR) was performed using two sets of primers targeting the mitochondrial cytochrome c oxidase subunit I (COI) gene: LCO1490 and HCO2198 (Folmer et al., 1994), amplifying a 658 bp fragment, and mlCOIintF (Leray et al., 2013) with jgHCO2198 (Geller et al., 2013), amplifying a 313 bp fragment. Each PCR reaction (25 µl) contained five µl of 5× HOT FIREPol® Blend Master Mix (Solis BioDyne, Tartu, Estonia), 0.5 µl of each primer (20 µM), one µl of DNA template, and 18 µl of nuclease-free water. PCR cycling conditions for LCO1490 and HCO2198 primers were as follows: initial denaturation at 95 °C for 15 min (hot-start for HOT FIREPol® Blend Master Mix); followed by 5 cycles of 94 °C for 30 s, 45 °C for 1 min, and 68 °C for 1 min; then 35 cycles of 94 °C for 30 s, 51 °C for 1 min, and 68 °C for 1 min; with a final extension at 68 °C for 5 min. For the mlCOIintF and jgHCO2198 primer pair, the PCR cycling conditions were as follows: initial denaturation at 95 °C for 15 min; 35 cycles of 95 °C for 30 s, 57 °C for 30 s, and 72 °C for 1 min; followed by a final extension at 72 °C for 10 min. Each sample was amplified in duplicate. Duplicates were pooled and PCR product yields were verified using gel electrophoresis. Based on the band intensity, the PCR products per sample were pooled into an amplicon library (per primer set) as follows: seven µl for weak bands (and for PCR blanks), three µl for moderate, and one µl for strong bands. Amplicon libraries were purified using the FavorPrep™ Gel/PCR Purification Kit (Favorgen Biotech, Vienna, Austria). The longer COI fragment (658 bp) was sequenced using the PacBio Sequel II platform (The Norwegian Sequencing Centre, Norway), and the shorter fragment (313 bp) was sequenced on the Illumina NovaSeq 6000 platform (2 × 250  bp; Novogen, Sacramento, CA, USA).

Bioinformatics on metabarcoding data

Metabarcoding sequencing data were processed using PipeCraft2 v1.0.0 (Anslan et al., 2017; https://pipecraft2-manual.readthedocs.io), employing the DADA2 pipeline (Callahan et al., 2016) as implemented within PipeCraft2, using default settings for PacBio (PacBioErrfun) and Illumina (loessErrfun) data. Amplicon sequence variants (ASVs) generated by DADA2 were clustered at 97% similarity (in vsearch; Rognes et al., 2016) using the “ASV to OTU” module. Additional post-clustering with default settings was performed using the LULU algorithm (Froslev et al., 2017) via the “LULU” module . Potential index-switching (tag-jumping) artifacts were filtered using the “filter tag-jumps” module, employing UNCROSS2 (Edgar, 2018), with default settings. OTUs containing stop codons and sequences shorter than 310 bp (Illumina) or 649 bp (PacBio) were removed using the “filter numts” module. Taxonomic assignment of OTUs was performed using the RDP Classifier (Wang et al., 2007) with the CO1-classifier v5.1.0 database (Porter & Hajibabaei, 2018). OTUs were assigned to the class Arachnida (including mites) when the bootstrap confidence value was at least 0.8. The raw sequencing data are deposited in European Nucleotide Archive (ENA) at EMBL-EBI under accession number PRJEB90911.

Statistics

Mite community composition derived from PacBio and Illumina datasets, obtained from both 0.2 and 2 g soil samples, were compared to each other (29 samples from five sites; Table 1) and a subset of samples (17 samples from three sites, Table 1) to the morphological identification dataset. Differences in OTU/morphospecies richness between methods were tested with a pairwise Wilcoxon rank-sum test (with Bonferroni correction) as implemented in the stats package v4.1.3 in R v4.1.3 (R-Core-Team, 2025). Overlap in mite family detections between morphological and metabarcoding datasets was visualized using an Euler diagram as implemented in the eulerr package v7.0.2 (Larsson, 2024). Ordinations of mite communities identified by different methods were compared using Procrustes analysis with the vegan package v2.6-10 (Oksanen et al., 2025). Ordinations were generated using Principal Coordinate Analysis (PCoA) from the ape package v5.7.1 (Paradis & Schliep, 2019). For ordination, data were transformed into presence/absence (1/0) matrices, and sample similarity was calculated using the Bray–Curtis index in vegan.

Results

DNA extraction from one sample from site2 (site2_6) failed; thus, is excluded from all datasets. The average sequencing depth (from 29 samples, across five sites; Table 1) per sample was 161,472 for Illumina and 2,142 for PacBio. In total, the Illumina data yielded 11,883 OTUs, of which 1,517 (12.8%) were classified as Metazoa (with bootstrap on ≥0.8; Table S1). The rest of the OTUs were classified as fungi, bacteria, plants, or remained unclassified at kingdom level. The PacBio dataset yielded 12,357 OTUs with 2,118 (17.1%) metazoan OTUs (Table S2). The number of mite OTUs was 59 and 61 for the whole (0.2 g + 2 g treatments) Illumina and PacBio datasets, respectively. In the Illumina dataset, the number of mite OTUs was 44 in the 0.2 g treatment and 38 in the 2 g treatment, whereas in the PacBio dataset, 36 and 40 OTUs in the 0.2 g and 2 g treatments, respectively. However, a pairwise Wilcoxon rank-sum test demonstrated no significant differences between any pair of treatments (all p > 0.3; Fig. S1). The same was found when analyzing all metazoan OTUs in the metabarcoding datasets (all p = 1; Fig. S2).

From the 17 samples subjected to Tullgren funnel extraction followed by morphological identification of mites (across three sites; Table 1), the Illumina dataset revealed 42 mite OTUs in the 0.2 g treatment and 25 OTUs in the 2 g treatment samples. PacBio dataset hosted 20 and 33 mite OTUs in 0.2 g and 2 g treatments, respectively. However, a significantly higher mite taxon richness (morphotypes) was identified in the morphological dataset (p < 0.001, Fig. 1). A total of 105 morphotypes, were identified based on morphological examination (Table S3).

Figure 1 Mite OTU/morphotypes richness.

Comparison of mite OTU/morphotypes richness detected using different identification methods (from sites 1 to 3, Table 1).

A total of 51 mite families were identified morphologically. In comparison, the Illumina dataset contained 22 families in total, with 13 families recovered from the 2 g and 21 from the 0.2 g treatments. PacBio dataset hosted 15 families overall, with 12 families detected in the 2 g treatment and 10 in the 0.2 g treatment. The Euler diagram (Fig. 2) illustrates the overlap and uniqueness of family-level detections among the different identification methods. The majority of families (32) were unique to the morphological dataset and not detected by either metabarcoding approach (Fig. 2).

Figure 2 Euler diagram showing the number of unique families detected by each method and overlap of mite families detected by different identification methods.

Sample replicates of 0.2 g and 2 g are pooled in metabarcoding datasets. Mite OTUs/morphotypes unclassified to the family level are excluded.

Tullgren funnel extraction, followed by morphological identification showed that all samples contained mites (Fig. 1). However, metabarcoding missed mites from 1 to 6 samples (depending on the identification method; Fig. 1). For comparisons of mite community structure ordinations, we included only those samples in which mites were detected in all treatments being compared. Despite the substantially higher mite richness observed in the morphological dataset, Procrustes analysis revealed a strong concordance in the relationships among samples, as represented in ordination space, between morphological and PacBio datasets (Procrustes R = 0.807, p < 0.001 for 2 g; Procrustes R = 0.763, p = 0.007 for 0.2 g; Fig. 3A). In contrast, the correspondence between morphological and Illumina datasets was much weaker and not statistically significant (Procrustes R = 0.399, p = 0.289 for 2 g; Procrustes R = 0.334, p = 0.496 for 0.2 g; Fig. 3B).

Figure 3 Comparison of mite community composition using Procrustes analysis of PCoA ordinations.

(A) Comparison between morphological (Morph.) and PacBio datasets (0.2 g and 2 g treatments); (B) comparison between morphological (Morph.) and Illumina datasets (0.2 g and 2 g treatments). Each point represents a sample in the first two principal coordinate axes. Lines connect each sample’s point in the morphological ordination (reference) to its corresponding point in the PacBio or Illumina ordination, illustrating the Procrustes residuals (the degree of mismatch between methods for each sample). Higher R values indicate greater concordance between methods.

Furthermore, we compared the relationships among samples from the PacBio and Illumina mite OTUs datasets. Procrustes analysis showed high and significant concordance between 0.2 g and 2 g treatments only within a metabarcoding dataset (Procrustes R = 0.887, p < 0.001 for PacBio 0.2 g vs. PacBio 2 g and Procrustes R = 0.859, p = 0.041 for Illumina 0.2 g vs. Illumina 2 g). There was a strong correlation also between PacBio 0.2 g and Illumina 0.2 g mite datasets (Procrustes R = 0.862), but this was marginally significant (p = 0.054). All other comparisons among the metabarcoding dataset revealed non-significant relationships among samples, as represented in ordination space (all p > 0.290 in Procrustes analyses). However, the analysis with all metazoan OTUs in the 0.2 g and 2 g PacBio and Illumina datasets (Tables S1 and S2) demonstrated strong concordance in the relationships among samples for all pairwise cases (Procrustes R > 0.923, p < 0.001). Similarly, when comparing metazoan OTUs datasets with the morphological data (only mites), strong concordance was also observed (Fig. 4; Procrustes R = 0.902, p < 0.001 between PacBio and morphological dataset; Procrustes R = 0.894, p < 0.001 between Illumina and morphological dataset).

Figure 4 Comparison of Metazoan OTU community compositions in metabarcoding datasets with mite morphospecies composition from the morphological dataset, using Procrustes analysis of PCoA ordinations.

Sample replicates of 0.2 g and 2 g are pooled per metabarcoding sample. Each point represents a sample in the space of the first two principal coordinate axes. Lines connect each sample’s position in the morphological ordination (reference) to its corresponding position in the PacBio or Illumina ordination.

Discussion

This study reveals insights into the effectiveness and limitations of DNA metabarcoding approaches when characterizing soil mite communities by using universal COI primers for generating amplicon libraries. Regardless of soil quantity input for DNA extraction, morphological identification of mite specimens revealed significantly higher richness across taxonomic levels compared with either metabarcoding approach. Despite that, our results demonstrated more consistent mite community composition patterns (in terms of sample similarities in an ordination space) between PacBio and morphological datasets, compared with Illumina and morphological datasets.

Although the morphological dataset contained higher mite richness, the metabarcoding approach detected additional taxa not identified by morphology, indicating complementary detection rather than a strict subset relationship. For example, species such as Steneotarsonemus laticeps (Halbert, 1923) and Sierraphytoptus ambulans (Chetverikov & Sukhareva, 2009) were only found in the PacBio dataset. Those species have never been recorded before in Estonia, but the closest records of S. ambulans are from Finland (Chetverikov & Sukhareva, 2009) and S. laticeps from Poland (Labanowski, Labanowska & Suski, 1990), suggesting rather under-sampling than erroneous metabarcoding records. It is known that some taxa may respond poorly to Tullgren funnel heat extraction (Coleman, Callaham & CrossleyJr, 2017), thus a complementary set of taxa from soil eDNA is expected (Young & Hebert, 2022).

Surprisingly, there were also substantial discrepancies in family-level diversity detected by different methods. Morphological identification revealed a total of 51 mite families, while metabarcoding approaches detected less than half this number (24 families for Illumina and 15 for PacBio; Fig. 2). Although the used COI primers aim to target metazoans, our metabarcoding datasets demonstrated that only 12–17% of all the OTUs were classified as Metazoa—an observation common in studies where universal COI primers are used to amplify soil eDNA (Anslan et al., 2021; Kirse et al., 2021; Watts et al., 2019; Young & Hebert, 2022). This is likely attributed to the diluted signal from soil mites, despite they are one of the most abundant groups of soil arthropods (Rosenberg et al., 2023). To increase the proportion of arthropod DNA for amplicon library generation, many studies extract the arthropods from soil samples prior to DNA extraction (Andujar et al., 2022; Arribas et al., 2021; Perdomo-González et al., 2025). However, when comparing results from morphological identifications of soil arthropods with metabarcoding extracted specimens, Basset et al. (2022) reported non-significant correlations between datasets, whereas other studies reported overall correlating richness and composition patterns between molecular and morphological approaches (Ustinova et al., 2021; Young & Hebert, 2022). Given the patchydistributions of soil arthropods (Bahram et al., 2016; Bardgett, 2002) the contrasting results from latter examples may stem from the different sampling methods: Basset et al. (2022) analyzed discrete paired samples (separated by ∼10 cm in the field), while Young & Hebert (2022) and Ustinova et al. (2021) homogenized the soil sample before extracting specimens and DNA. In our study, even with homogenized soil samples, morphological identification yielded considerably higher mite richness than either metabarcoding approach. While Oliverio et al. (2018) found similar results for mites, their study showed more consistent patterns between metabarcoding and morphological datasets for the entire arthropod community. This indicates that when a focus is on a specific soil arthropod group, such as mites, rather than the broader community patterns, methodological biases may become more pronounced. A similar observation has been noted, for example, in metabarcoding ostracods directly from lake sediment eDNA samples using universal COI primers (Echeverría-Galindo et al., 2021).

Despite detecting fewer mite taxa overall, the mite OTUs communities in PacBio dataset showed relatively strong concordance with morphological data in terms of the relationships among samples (Fig. 3A). Although the Illumina dataset had substantially higher sequencing depth and slightly higher mite Family richness, the Illumina dataset failed to align with the morphological data. This suggests that while both molecular methods are missing mite taxa, the longer reads generated may provide more reliable information for determining the community patterns. The better performance of long reads may not be attributed directly to the better performance of LCO1490/HCO2189 primers compared with mlCOIintF/jgHCO2198 as the overall mite OTUs richness was not significantly different between metabarcoding datasets. However, longer reads are less susceptible to amplifying relic, fragmented DNA (Taberlet et al., 2018). Consequently, the community patterns from long reads may reveal better representation of the active fauna, which may explain the stronger alignment with the morphological data derived from active mite extraction.

Within a set of 17 samples from three sites (used for the morphological comparison; Table 1), the Illumina 0.2 g dataset detected significantly more mite OTUs than the 2 g dataset. This is counterintuitive, as one would typically expect larger soil volumes to yield greater taxonomic richness, especially of metazoans, as demonstrated in previous studies (Kirse et al., 2021). While herein used PowerMax Soil Kit is designed for larger starting volumes, its protocol suggests eluting the DNA into a relatively large volume (up to 10 mL) compared to the PowerSoil Pro Kit, which uses a much smaller elution volume (up to 100 µl). This large elution volume in the PowerMax kit may result in a final DNA extract that is more diluted. Indeed, in our study, the average DNA concentration from the 2 g samples (PowerMax Kit) was 4.85 ng/µl, whereas the 0.2 g samples (PowerSoil Pro Kit) yielded an average concentration of 8.48 ng/µl. We hypothesize that this lower DNA concentration from the 2 g samples served as a less effective template during PCR amplification, leading to the detection of fewer mite OTUs. However, varying the initial soil amount for DNA extraction (0.2 g vs. 2 g) did not result in a statistically significant difference in mite OTU richness for either the Illumina or PacBio metabarcoding approaches across the 29 samples from 5 sites (Fig. S1). Despite that, the mite OTUs composition as represented in the ordination space significantly correlated only within each metabarcoding dataset (i.e., 0.2 g and 2 g treatments significantly correlated within PacBio or Illumina datasets) but not between them. This indicates that the choice of metabarcoding method (herein PacBio vs. Illumina) had a greater influence on the detected mite community composition than the amount of starting material. However, the importance of sample quantity may be critical at larger scales, as other studies have found that input material substantially greater than 2 g yields higher metazoan richness and significantly different community compositions compared with low inputs such as 0.2 g (Nascimento et al., 2018) or 0.5 g (Kirse et al., 2021).

In contrast to the results for mites alone, the analysis of the entire metazoan community showed high concordance across all metabarcoding treatments. Both community composition and total OTU richness were consistent across all treatments. A particularly interesting finding was the strong Procrustes correlation between the morphological mite data and the entire metazoan community from both metabarcoding datasets (Fig. 4). This indicates that soil mites serve as a powerful ecological indicator group (Gulvik, 2007), where the community structure of mites alone mirrors the broader patterns of the entire soil metazoan community. The environmental parameters shaping the mite assemblages appear to be similar to those for other soil animal groups, driving their structure. It also demonstrates that even very small quantities of soil used generally for soil microbial analyses can yield meaningful data about soil faunal communities. This validates the use of broad-spectrum metabarcoding for biomonitoring, as it suggests that whole community metabarcoding data can effectively reveal key ecological patterns across the landscape, much like those captured using traditional indicator species.

Conclusions

In conclusion, our study demonstrates that while soil eDNA metabarcoding with universal COI primers is a powerful tool, it has significant limitations for characterizing a specific taxonomic group like soil mites, capturing less than identified through morphological analysis. This suggests that for studies focusing on a specific faunal group, a preliminary extraction of the organisms from the soil is advisable which allows for the processing of a much larger soil volume than is typically feasible for direct DNA extraction, thereby increasing the likelihood of capturing a more comprehensive profile of the specific community of interest. But when working with soil eDNA, our results indicate that methodological choices (PacBio vs. Illumina) have a greater impact on mite community composition than simply input soil quantities (0.2 g, 2 g) for DNA extraction. The long-read PacBio dataset provided a more ecologically reliable community profile that aligned strongly with the morphological data. Finally, the patterns of the entire metazoan community strongly mirrored those of the morphologically identified mite communities alone. This validates the role of mites as powerful ecological indicators and demonstrates that broad-spectrum metabarcoding is a highly effective tool for large-scale biomonitoring, capable of revealing key ecological patterns even without perfect taxonomic resolution of every single group.

Supplemental Information

Supplemental Information 1 Illumina dataset

Supplemental Information 2 PacBio dataset of Metazoan OTUs, including mite OTUs

The prefix “b” in the sample header indicates samples subjected to the 2 g DNA extraction treatment, while “s” denotes the 0.2 g DNA extraction treatment. The numbers following the prefix represent the site number (first digit) and the replicate number in that site along the transect (second digit). Column ”*_BootS” denotes the RDP bootstrap value for the corresponding taxonomic level.

Supplemental Information 3 Morph dataset

Supplemental Information 4 Comparison of mite OTU richness detected using PacBio and Illumina datasets

Supplemental Information 5 Comparison of metazoan OTU richness detected using PacBio and Illumina datasets across five sites (Table 1)

The authors acknowledge CSC–IT Center for Science, Finland, for computational resources, Novogene and Norwegian Sequencing Centre (NSC) for sequencing services.

Additional Information and Declarations

Competing Interests

Author Contributions

Data Availability

The autors declare no competing interests.

Sirle Varusk conceived and designed the experiments, performed the experiments, analyzed the data, prepared figures and/or tables, authored or reviewed drafts of the article, and approved the final draft.

Kaarel Sammet conceived and designed the experiments, performed the experiments, authored or reviewed drafts of the article, and approved the final draft.

Manikandan Ariyan performed the experiments, authored or reviewed drafts of the article, and approved the final draft.

Khairiah Mubarak Alwutayd analyzed the data, prepared figures and/or tables, authored or reviewed drafts of the article, and approved the final draft.

Sten Anslan conceived and designed the experiments, analyzed the data, prepared figures and/or tables, authored or reviewed drafts of the article, and approved the final draft.

The following information was supplied regarding data availability:

The raw sequencing data are available at the European Nucleotide Archive (ENA) EMBL-EBI: PRJEB90911.

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
