# Peer review of "DNA metabarcoding of mites from small soil samples: limited agreement with morphological identifications but improved results from long-read sequencing"

_PeerJ, doi:10.7717/peerj.20205_

## Round 0.1 · original submission · Major Revisions

Dear Dr. Anslan, I ask you to respond to all the reviewers' fundamental comments and hope that the new version of this manuscript will be approved by them for publication.

·

Basic reporting

Dear authors,
I have carefully read the manuscript DNA metabarcoding of mites from small soil samples: limited agreement with morphological identifications but improved results from long-read sequencing. The manuscript analyzes the congruence of soil mite communities between morphological and metabarcoding approaches using the mitochondrial gene COI. The introduction and background showed enough context, and the literature is well reference and relevant. Also, the manuscript is well-written and easy to follow. However, there are some typos that need to be corrected which I pointed them out in the general comments section.

Experimental design

The research question is well-established, and it is fitted under the scope of Peerj. In addition, the methods are described with sufficient detail and information to replicate, please find the specific comments in the additional comments section. Also, the literature gap is filled with relevant information. I only suggested to include more information about the Illumina sequencing process, regarding the use and sequencing of the paired-end reads (R1 & R2), in this case the authors used the paired-end reads which are shown in the databases, but not in the Materials & Methods section, please find my comment in the additional comments section.

Validity of the findings

The research design is appropriate, and the results are clearly presented. All the underlying data is well-represented in Table 1 and across the text. Also, the raw data is well described in public databases. I only would like to suggest a better format of Table 1, where for example the head of the column “coordinates” fits completely in the cell.

Additional comments

L71-74: It is not clear if the previous studies are (Doorenspleet, Latz & Srivathsan)? Or on the contrary are new studies? Could you provide the references of those studies?
L135: Check the description in this line, Illumina NovaSeq 6000 platform paired-end reads (2 x 313 bp)?
L103-L108: The identification of Oribatida and mesostigamata were assessed following a taxonomic key?
L155: The correct word is Statistics
L234-237: Add the Authority name after the species name
L255: The correct word is composition patterns
L422: Check the journal in this line, in agreement with the other references should be Frontiers in Zoology
L449: Same as above for Sci Adv.
In figure 4 what you mean by (reference)?
In Table 1, the word is dominated and Salix sp. (sp. without italics)

Reviewer 2 ·

Basic reporting

In this study, the authors applied two approaches to assess the taxonomic identification of mites from soil samples, one through morphological identification and the other through metabarcoding. In general, the article is properly written and the English level is fine, although some improvements can be made as to make the reading smoother and more straightforward. The aim of the research and the hypothesis are clear, and the procedures and results address properly the scientific questions. The article structure is correct as are the figures and tables. The raw data is as well available in the database as indicated by the authors. Nonetheless, some points should be checked before accepting the article for publication. First, even if sufficient field background is provided, several parts of the introduction and the discussion sections lack of proper literature references. Besides, some sentences/paragraphs are wrongly allocated to the article sections (i.e., materials and methods information written in the introduction, or results in the discussion). Apart from this, the material and methods sections are imbalanced: the metabarcoding approach is extensively described, but the morphological identification is poorly detailed and many important information is missing, such as taxonomic keys used for the identification or credits to the taxonomist that performed the ID. Improving this aspect might give a stronger support to the results. Finally, the results section may benefit from a tidier organization, maybe dividing it into different paragraphs according to the approaches / treatments. In some cases, also the statistical results are wrongly reported (i.e., missing statistic value, df, number of replicates).

Experimental design

The experimental design looks correct and the number of replicates per treatment sufficient for the purposes of the study.

Validity of the findings

No comment.

Additional comments

Please find in the attached pdf my improvement suggestions per article section / line.

Annotated reviews are not available for download in order to protect the identity of reviewers who chose to remain anonymous.

·

Basic reporting

See below

Experimental design

see below

Validity of the findings

The manuscript primarily compares the number of OTUs generated from Illumina and PacBio sequencing and relates these to morphological findings. However, it lacks sufficient taxonomic depth. The authors claim that PacBio sequencing results closely match morphological outcomes, but comparisons were made mainly at the family level using OTUs derived from Illumina and PacBio data. This level of comparison is inadequate for robust validation. To ensure confidence in the outcomes relative to morphological identifications, comparisons should be extended to the genus or species level. While the authors have acknowledged some limitations, these do not adequately address the core analytical gaps. Overall, the manuscript lacks in-depth taxonomic resolution, which is a significant flaw that must be addressed before further review.

Line 186–187: The Illumina dataset revealed 45 mite OTUs in the 0.2 g treatment and 26 OTUs in the 2 g treatment. Please explain why a tenfold reduction in soil sample size resulted in nearly double the number of OTUs.

Line 195–196: The manuscript notes that 32 families were unique to the morphological dataset and not detected by either metabarcoding approach. Please also include the reverse—families detected exclusively by metabarcoding and not found in the morphological dataset.

Line 234: The exotic species Steneotarsonemus laticeps and Sierraphytoptus ambulans were found only in the PacBio dataset. Since the manuscript discusses 'exotic incursions', please clarify whether their presence was confirmed via PCR.

Line 51: “51 mite families were identified via morphology”—the authors should provide genus and species-level details, including the number of species identified and their respective genera.

---

## Round 0.2 · accepted · Accept

Dear Dr. Anslan, I congratulate you on the acceptance of this article for publication and hope that you will continue to publish such interesting manuscripts in our and other scientific journals.

·

Basic reporting

Dear Authors,
After reviewing the previous comments, I believe the manuscript has improved, and from my side I have no further remarks to add.

Experimental design

Dear Authors,
No comments to add.

Validity of the findings

Dear Authors,
No comments to add.

Additional comments

Dear Authors,
No comments to add.

Reviewer 2 ·

Basic reporting

Authors have adressed all points properly.

Experimental design

(no comment)

Validity of the findings

(no comment)

Additional comments

(no comment)